# Peripheral Nerve Regeneration–Adipose-Tissue-Derived Stem Cells Differentiated by a Three-Step Protocol Promote Neurite Elongation via NGF Secretion

**DOI:** 10.3390/cells11182887

**Published:** 2022-09-15

**Authors:** Silvan Klein, Andreas Siegmund, Andreas Eigenberger, Valerie Hartmann, Felix Langewost, Nicolas Hammer, Alexandra Anker, Konstantin Klein, Christian Morsczeck, Lukas Prantl, Oliver Felthaus

**Affiliations:** 1Department of Plastic, Hand and Reconstructive Surgery, University Medical Center Regensburg, Franz-Josef-Strauss-Allee 11, 93053 Regensburg, Germany; 2Department of Radiology, University of Cologne, Kerpener Str. 61, 50937 Cologne, Germany; 3Department of Cranio- and Maxillofacial Surgery, University Medical Center Regensburg, Franz-Josef-Strauss-Allee 11, 93053 Regensburg, Germany

**Keywords:** adipose-tissue-derived stem cells, Schwann cells, nerve regeneration, neural differentiation, neurotrophic factors

## Abstract

The lack of supportive Schwann cells in segmental nerve lesions seems to be one cornerstone for the problem of insufficient nerve regeneration. Lately, adipose-tissue-derived stem cells (ASCs) differentiated towards SC (Schwann cell)-like cells seem to fulfill some of the needs for ameliorated nerve recovery. In this study, three differentiation protocols were investigated for their ability to differentiate ASCs from rats into specialized SC phenotypes. The differentiated ASCs (dASCs) were compared for their expressions of neurotrophins (NGF, GDNF, BDNF), myelin markers (MBP, P0), as well as glial-marker proteins (S100, GFAP) by RT-PCR, ELISA, and Western blot. Additionally, the influence of the medium conditioned by dASCs on a neuron-like cell line was evaluated. The dASCs were highly diverse in their expression profiles. One protocol yielded relatively high expression rates of neurotrophins, whereas another protocol induced myelin-marker expression. These results were reproducible when the ASCs were differentiated on surfaces potentially used for nerve guidance conduits. The NGF secretion affected the neurite outgrowth significantly. It remains uncertain what features of these SC-like cells contribute the most to adequate functional recovery during the different phases of nerve recovery. Nevertheless, therapeutic applications should consider these diverse phenotypes as a potential approach for stem-cell-based nerve-injury treatment.

## 1. Introduction

Schwann cells (SCs) have been identified as the major regenerative source of the peripheral nervous system [1,2]. The unique growth-promoting properties of SCs within nerve lesion sites allow signaling, as well as the coordination of regenerative processes, ranging from the spinal-cord level to effector organs. During the course of nerve injury, SCs dedifferentiate in response to the injury into a state that has been termed the activated SC phenotype [3]. SC activation seems to upregulate a distinct regenerative transcriptional program that creates a permissive environment for nerve regeneration [4,5]. With regard to the reconstruction of segmental nerve injuries, promising results were achieved by both the transplantation of SCs and mesenchymal stem cells into injury sites [6,7,8,9,10,11,12].

Unlike SCs, adipose-tissue-derived stem cells (ASCs) can be harvested in great numbers with minimally invasive methods. Furthermore, the ASCs differentiated into SC-like cells yielded a significant improvement in the functional restitution in previous in vivo investigations [6,7,8,11,12]. Remarkably, the regenerative potential of ASCs was observed locally at the level of the nerve lesion, as well as at the spinal-cord level, which suggests the capability of ASCs to secrete growth factors that are comparable to SCs [6,11].

Although various studies have applied SC-like ASCs for the purpose of peripheral nerve regeneration, precise phenotypes of SC-like ASCs have found little attention yet. In this study, various culture conditions were investigated for their ability to efficiently direct ASCs into specialized SC phenotypes. Three distinct differentiation protocols were investigated for their ability to induce the expressions of specific glial cell markers, as well as the secretion of neurotropic and neurotrophic factors, both at the transcriptional and translational levels. In order to transfer these data, the results were reproduced on surfaces that could be used as nerve guidance conduits. These artificial tubes are used for guiding axonal regrowth, and they can be manufactured from a variety of materials, ranging from simple proteins, such as extracellular matrix components (collagen, fibronectin, laminin), to complex materials, such as spider silk. In their simplest implementation, these conduits are simply bridging the gap in the critical-size nerve defects, giving the axon the right direction, and protecting the regeneration site from interfering substances. However, they can also be bioengineered to release growth factors, or could be used as a scaffold for the introduction of regenerative cells [9].

Furthermore, indirect coculture experiments were conducted to verify the potential of the differentiated ASCs (dASCs) to promote the in vitro axonal regeneration in a neuroblastoma cell line.

## 2. Materials and Methods

### 2.1. ASC Cell Culture

ASCs were isolated as previously described [9]. Briefly, freshly harvested adipose tissues from the inguinal fat pads of rats were minced into small pieces (1 mm^3^), and were subsequently digested with 2U collagenase (Sigma Aldrich, St. Louis, MO, USA) per mL of fat tissue at 37° C for 60 min. Afterwards, the cell suspension was passed through a 100 µm filter (MERCK Millipore, Billerica, MA, USA) and was centrifuged for 5 min at 500 *g*. The cell pellet was washed, the cells were seeded into T175-cell-culture flasks (Greiner Bio One, Kremsmünster, Austria), and they were allowed to adhere in Eagle’s minimum essential medium (MEM) (Sigma Aldrich), supplemented with 20% fetal bovine serum (Pan-Biotech, Aidenbach, Germany) and a penicillin–streptomycin solution (Sigma Aldrich) (referred to as the proliferation medium below). Blood cells and other nonplastic-adherent cells were removed by washing with phosphate-buffered saline (PBS) (Sigma Aldrich). Upon reaching subconfluency, the cells were detached from the surface using a trypsin/EDTA (ethylenediaminetetraacetic acid) solution (Pan-Biotech). The cells in Passage 5 were used for all the experiments.

### 2.2. Differentiation Protocols

We chose three different protocols for the neural differentiation from the literature, which differ in terms of the duration, surface coating, and media composition. The original protocols are as follows. Protocol 1 (P#1): Ham’s F12 (Sigma Aldrich) supplemented with 20 ng/mL epidermal growth factor (EGF) (PeproTech) and 20 ng/mL basic fibroblast growth factor (bFGF) (PeproTech, Cranbury, NJ, USA), until the cells detached as neurospheres from the surface. These neurospheres were transferred into poly-L-lysin-coated well plates, in which the spheres attach, and an outgrowth of the cells was observed for an additional 48 h in proliferation medium supplemented with 5 µM tretinoin (all-trans-retinoic acid), 252 ng/mL neuregulin 1, 5 ng/mL platelet-derived growth factor (PDGF), and 14 μM forskolin [13]. Protocol 2 (P#2): Proliferation medium supplemented with N2 Supplement (Life Technologies, Carlsbad, CA, USA), 20 ng/mL bFGF, 20 ng/mL EGF, 5 µM tretinoin, and 5 ng/mL transforming growth factor beta 1 (TGF-beta1) for five days [14]. Protocol 3 (P#3): After reaching subconfluency, cells were incubated with proliferation medium supplemented with 1 mM beta-mercaptoethanol (Sigma Aldrich) for 24 h. After that, the medium was changed to proliferation medium supplemented with 35 ng/mL tretinoin for 72 h. Subsequently, the cells were incubated in proliferation medium supplemented with 5 ng/mL PDGF, 10 ng/mL bFGF, 14 μM forskolin, and 252 ng/mL neuregulin 1 for two more weeks [15].

To test whether the effects of these differentiation protocols could be enhanced, the three basic protocols were additionally supplemented with a variety of growth factors and compounds also related to neural differentiation in order to optimize the glial/Schwann-cell-like differentiation. The tested growth factors were comprised of tacrolimus (FK-506), forskolin, growth-arrest-specific gene-6 (GAS-6), heregulin beta 1, melatonin, neuregulin 1, PDGF, and TGF-beta 1. All tested supplements were tested as an addition in each of the three basic protocols, unless they were already part of the original differentiation medium. The protocols and supplements were tested in various combinations and concentrations, and they were evaluated for their capacity to optimize the differentiation in ASCs. However, data are shown for the most effective supplement in each of the three protocols only. These are Protocol 1, supplemented additionally with 5 ng/mL of TGF-beta1; Protocol 2, supplemented additionally with 252 ng/mL of neuregulin 1; Protocol 3, supplemented additionally with 5 ng/mL of TGF-beta1. The protocols are summarized in Table 1. Pictures were taken with the Wilovert S microscope (Helmut Hund GmbH, Wetzlar, Germany) and the ScopeTec DCM 800 camera, using the Scope Photo 3.0 software.

### 2.3. NG108-15 Cell Culture

The NG108-15 (mouse neuroblastoma × rat glioma hybrid) cell line was purchased from the ECACC (European Collection of Authenticated Cell Cultures) (Salisbury, UK), and it was cultured in a proliferation medium until the coculture experiments started.

### 2.4. Coculture

The ASCs were differentiated as described above. On the second but last day of the respective differentiation protocols, the medium was changed and allowed to be conditioned for 24 h. Subsequently, the medium was collected and used in indirect coculture experiments. The NG108-15 cells cultured in conditioned media were used for the RNA isolation or neurite-outgrowth assay.

### 2.5. RNA Isolation/Reverse Transcription PCR

The total RNA from differentiated ASCs and their respective controls, and from NG108-15 cells cultured with dASC-conditioned medium, was isolated using the RNeasy Mini Kit (Qiagen, Hilden, Germany), according to the manufacturer’s instructions. Isolated RNA was reversely transcribed using the QuantiTect Reverse Transcription Kit (Qiagen). Real-time RT-PCR was performed using the DyNAmo HS SYBR Green qPCR Kit (Life Technologies) and the Eco™ Real-Time PCR System (Illumina, San Diego, CA, USA). Primers were obtained from Eurofins MWG Operon. Primer sequences are listed in Table 2. The relative gene expressions were calculated using the ΔΔCt method, and they were normalized to an undifferentiated control using glycerinaldehyd-3-phosphat-dehydrogenase (GAPDH) as the housekeeping gene. The gene expression was evaluated for genes related to neural differentiation (angiotensin II receptor type 2 (AGTR2), brain-derived neurotrophic factor (BDNF), glial cell line-derived neurotrophic factor (GDNF), glial fibrillary acidic protein (GFAP), myelin basic protein (MBP), nerve growth factor (NGF), myelin protein zero (P0), low-affinity nerve growth factor receptor (p75), calcium-binding protein B (S100B), and TGF beta receptor (TGFβR)). Samples were evaluated in biological quadruplicates.

### 2.6. Western Blot

The total protein was isolated using RIPA cell lysis buffer supplemented with protease inhibitors (Complete Mini, Roche, Basel, Switzerland). SDS-PAGE was performed using the Mini-PROTEAN^®^ Tetra Cell electrophoresis chamber (BioRad, Hercules, CA, USA). The PageRuler Prestained NIR Protein Ladder (Thermo Fisher Scientific) was used. Separated proteins were blotted onto a nitrocellulose membrane (Amersham Protran 0.2 NC, Sigma-Aldrich) using the Mini Trans-Blot^®^ Cell wet blotting system (BioRad). Membranes were incubated with primary antibodies (rabbit anti-rat p75 NGF receptor antibody; mouse anti-rat beta-actin antibody (both Abcam, Cambridge, UK)) overnight at 4 °C, and infrared-labeled secondary antibodies (IRDye 800CW donkey anti-goat, Li-Cor; IRDye 680CW goat anti-mouse (Li-Cor Biosciences, Lincoln, NE, USA)) for one hour at room temperature. Visualization was performed using the Odyssey^®^ Infrared Imaging System.

### 2.7. ELISA

For the ELISA, the cell-culture supernatant was collected during the differentiation processes. The same differentiation medium not conditioned by cells was used as the control. The rat NGF Duo Set ELISA Kit (R&D Systems, Minneapolis, MN, USA) and the GDNF (Rat) ELISA Kit (Abnova, Taipei, Taiwan) were used, according to the manufacturers’ instructions. Absorbance was measured using the VarioScan (ScanLab Puchheim, Germany) plate reader.

### 2.8. Neurite-Outgrowth Assay

For the neurite-outgrowth experiments, NG108-15 cells were seeded at a density of 800 cells/cm^2^, and they were allowed to adhere. Subsequently, the medium was changed to the media conditioned by dASCs. A differentiation medium that had not been conditioned was used as the control. After 48 h, pictures were taken, as described above. In order to verify the results, NG108-15 cells were also incubated with either 100 ng/mL recombinant NGF (PeproTech) or a 1:20,000 dilution of the neutralizing anti-NGF antibody (Sigma-Aldrich). For the comparison of the neurite numbers and neurite lengths between the three protocols, approximately 100 cells (up to 10 cells for at least 10 images) with varying numbers of neurites were measured for each condition. For the measurement of the neurite length after the NGF supplementation or NGF inhibition, 150 cells with varying numbers of neurites from three independent experiments were evaluated. FIJI software (based on ImageJ 1.53c, https://imagej.nih.gov/; https://fiji.sc/) was used for the neurite-outgrowth assessment.

### 2.9. Surface Coating

Cell-culture surfaces were coated with ECM proteins or poly(amino acids), as previously described [16]. Briefly, collagen I solution from bovine skin (10 µg/cm^2^), fibronectin from bovine plasma (5 µg/cm^2^), laminin from murine sarcoma basement membrane (1 µg/cm^2^), poly-L-lysine solution (3 µg/cm^2^), and poly-L-ornithine solution (2 µg/cm^2^) (all Sigma Aldrich) were prepared with PBS in the mentioned concentration. A total of 2 mL of the respective solutions was placed into each well of 6-well plates. For the poly-L-lysine and poly-L-ornithine solutions, coating was allowed for five minutes, whereas the coating with collagen I, fibronectin, and laminin endured several hours. After incubation, the coating solution was discarded, and the wells were washed with ddH_2_O and air-dried. For the glial differentiation on the coated surfaces, nothing was changed for Protocols 2 and 3, except the initial seeding into the coated wells. In Protocol 1, which already included the transfer of detached neurospheres into poly-L-lysine-coated wells, the cells were initially seeded into uncoated wells, and the poly-L-lysine was replaced with the respective coating material for the neurosphere transfer.

### 2.10. Statistical Analysis

A Student’s *t*-test was utilized for the assessment of the significance. In all diagrams, the mean values and either standard deviations or standard errors of the means are shown. A Shapiro–Wilk test was used to test for normal distribution.

## 3. Results

### 3.1. Morphological Appearance

Differentiation with Protocol 1 leads to a neurosphere morphology (Figure 1a). ASCs differentiated with Protocol 2 tend to aggregate into clusters (Figure 1b). Treatment with Protocol 3 (Figure 1c) showed the least impact on the ASC morphology compared with the undifferentiated control (Figure 1d).

### 3.2. RT-PCR after Differentiation of ASCs with Basic Protocols

The differentiation of the ASCs with Protocol 1 led to an upregulation of GFAP and S100B. While GFAP was upregulated 8-fold, S100B was upregulated 25-fold. MBP was upregulated 4.5-fold. Incubating the cells with Protocol 2 resulted in a higher expression of MBP, which was upregulated 6-fold, and a 120-fold upregulation of P0. p75 was upregulated 10-fold. Differentiation with Protocol 3 led to elevated expressions of GDNF, BDNF, and NGF, which were upregulated 6-, 15-, and 50-fold, respectively (Figure 2).

### 3.3. Western Blot after Differentiation of ASCs with Basic Protocols

The expression of p75 on the protein level was detected after differentiation with Protocol 2 only (Figure 3).

### 3.4. ELISA after Differentiation with Basic Protocols

High levels of NGF were detected after differentiation with Protocol 3 only (Figure 4).

### 3.5. RT-PCR after Differentiation of ASCs on Coated Surfaces

Selected marker genes were tested for differential regulation, in which their respective differentiation protocols were conducted on the coated surfaces. The results on the coated surfaces were calibrated to the basic protocol, in which the respective upregulation was observed. Compared with the basic Protocol 1, the GFAP upregulation was not impaired on the collagen surface, and it even increased on the fibronectin surface. However, on laminin and poly-L-ornithine, the GFAP upregulation was slightly impaired (Figure 5). The observed upregulation of MBP and P0 after differentiation with Protocol 2 was reduced on most of the coated surfaces. Both the collagen and fibronectin coatings led to a slight decrease in the MBP upregulation and P0 upregulation. The laminin coating caused a minor decrease in P0 upregulation, and a major decrease in MBP upregulation. The upregulations of both MBP and P0 were nearly unaffected by the poly-L-lysine coating, whereas the poly-L-ornithine coating led to a major decrease in the MBP upregulation, and the expression of P0 was unaffected (Figure 5). The upregulation of NGF after differentiation with Protocol 3 was not impaired on any of the surfaces. On the laminin, however, a significant increase was observed (Figure 5).

### 3.6. ELISA after Differentiation on Coated Surfaces

The upregulation of NGF after differentiation with Protocol 3 was reproduced on all the coated surfaces (Figure 6). However, the NGF secretion after differentiation on the laminin was only slightly increased compared with the uncoated control.

### 3.7. Neurite-Outgrowth Assay

The NG108-15 cells were growing in the differentiation media for Protocols 1, 2, and 3, with and without being conditioned by dASCs (Figure 7). Whereas the conditioning of the media had no impact on the neurite number (Figure 8), the mean neurite length significantly increased when the dASCs were allowed to secrete neurotrophins into the medium for 24 h (Figure 9), showing the effect of the dASCs differentiated with Protocol 3 on the neuron-like cell neurite outgrowth. The supplementation of both DMEM and the unconditioned Protocol-3 medium with NGF had a similar effect on the neurite outgrowth as dASC conditioning after predifferentiation with Protocol 3 (Figure 10). The addition of an NGF-neutralizing antibody reduced the effect of the conditioned medium on the neurite outgrowth significantly (Figure 11). This is in accordance with the observed upregulation of NGF in Protocol 3.

### 3.8. RT-PCR from NG108-15 Cells after Cultivation with Conditioned Media

The coculture of the NG108-15 cells with medium conditioned with Protocol-3-differentiated ASCs led to upregulations of p75, TGFbeta receptor, and angiotensin II type 2 receptor (Figure 12), although statistical significance was narrowly missed.

## 4. Discussion

It was shown that the acellular nerve scaffolds seeded with bone-marrow-derived stem cells (BMSCs) promoted and supported axon regeneration. Recently, it was shown that ASCs, which can be harvested in greater numbers with minimally invasive procedures, are a promising alternative [17,18,19]. SC activation is a physiological process in reaction to severe nerve trauma that includes a loss of myelinating proteins and, at the same time, allows for the release of the neurotrophic and neurotropic factors in SCs [3,20,21,22,23]. Originally, this distinct dedifferentiation was believed to imply a reversion into an earlier developmental stage in the SC lineage [24]. Lately, it has been hypothesized that the activated SC phenotype shows a closer relation to embryonic stem cells as opposed to neural crest cells, which might be one explanation for the favorable results that have been reported in nerve regeneration with various stem-cell sources [25,26]. However, for complete nerve regeneration, an activated phenotype with neurotrophin secretion, as well as a myelinating phenotype, seem to be necessary.

With regard to peripheral nerve regeneration, classic neurotrophins, such as NGF, and BDNF, as well as the neurotropic factor GDNF, have received the greatest attention because of their strong potential to promote posttraumatic neuronal survival and axonal regeneration [19,27,28,29,30,31]. In this study, the ASCs were subjected to three distinct differentiation conditions to compare the induced SC-like phenotypes with respect to their expression profiles of neurotrophic factors and myelin-forming proteins. The highest levels of NGF, GDNF and BDNF expressions were detected in the ASCs after differentiation with Protocol 3. Negativity for GFAP and positivity for P0 are distinct features of myelinating SCs, and they have been attributed to substances that induce intracellular cAMP elevation, such as forskolin [32]. This expression pattern was detected in the ASCs after differentiation with Protocol 2. The ASCs after differentiation with Protocol 1 showed elevated expressions of the general glial markers. Correspondingly, differences in the morphological appearance of the dASCs following the differentiation were observed. dASCs expressing myelin genes tend to form multicellular clusters, resembling the myelin sheath of axons, whereas ASCs resembling the activated dedifferentiated phenotype with regard to the gene expression morphologically resemble undifferentiated ASCs.

Xu et al. (basis for Protocol 1) applied EGF and bFGF to create neurospheres from ASCs, and they then cultured these cells with tretinoin, neuregulin 1, and PDGF to differentiate the ASCs into SC-like cells [13]. The corresponding dASCs showed myelinating capacities in vitro when cocultured with PC12 cells (i.e., a cell line of pheochromocytoma in rats), and they further induced neurite outgrowth in SH-SY5Y cells. Remarkably, although S100 was significantly upregulated in all of the investigated protocols, the strongest expression of this SC marker was detected with this differentiation. This group concluded that the dASCs of this protocol resemble the SC morphology, phenotype, and functional capacities. MBP was significantly elevated in this protocol, which supports the myelinating capacities that were reported in a coculture with PC12 cells [13]. This is in accordance with the observations of SC behavior under similar conditions [16].

Remarkably, when Dai et al. investigated the potential of ASCs for the regeneration of peripheral nerves with cell-seeded nerve conduits in vivo, the most promising results were found for the combination of undifferentiated ASCs and SCs [6]. This group suggested high expression rates of NGF in in vitro cocultures of ASCs, and the myelinating properties of SCs as a possible mechanism for this promising observation [6]. These reports are in line with the hypothesis of the neuromodulatory paracrine effects of ASCs, which might enhance in vivo nerve regeneration independently from differentiation towards SC-like phenotypes [9,15]. In SCs, the expression of neurotrophins is accompanied by a corresponding downregulation of P0 and MBP [4,33]. This myelin clearance is believed to be crucial for the formation of the bands of Büngner that guide sprouting axons across nerve lesions to ultimately reinnervate the original target organ [24]. Nevertheless, for the completion of the regenerative process, SCs again upregulate such myelin-forming proteins to rebuild the myelin sheath for the latter’s rapid conduction of action potentials [4,34]. However, this change in the myelin-forming-protein expression has hardly been considered in the field of stem-cell-based nerve regeneration, and it might be one reason for the controversial results after the transplantation of SC-like dASCs in previous publications [6,10,35,36,37,38,39].

Laminin is a component of the basement membrane, and it has been identified as a crucial factor for the regulation of SC signaling [40]. Interestingly, laminin was the only coating material to increase the NGF transcription of dASCs with regard to the investigated culture conditions, although the effect was small on the protein level. However, this effect was restricted to dASCs that lacked P0. In contrast, laminin does not seem to promote the secretion of NGF in dASCs phenotypes, which resemble myelinating SCs [41].

Neurite outgrowth in a neuroblastoma cell line (NG108-15) was observed especially after indirect coculture with the Protocol-3-differentiated ASCs. The supplementation of the unconditioned medium with NGF had a similar effect, whereas an NGF-inhibiting antibody mitigated the effect of the Protocol-3-conditioned media, suggesting the important role of NGF in axon elongation, and therefore in initial nerve regeneration. Whether a higher antibody concentration could have reduced the neurite outgrowth to the values seen for the unconditioned, or whether other factors in the conditioned medium contribute strongly to neurite elongation, remain to be evaluated. However, the gene-expression changes in the NG108-15 cells imply the latter. Neurite outgrowth is known to be associated with TGFβ and is induced by AGTR2 [42]. After culturing with Protocol-3-conditioned medium, the expressions of AGTR2 and TGFβ were upregulated significantly in the NG108-15 cells, but not in the NGF-supplemented medium. This indicates that Protocol-3-differentiated ASCs secrete other factors that are important for neurite elongation. Additionally, the expression of neurotrophin receptor p75 was upregulated too.

The SC-like cells from differently treated ASCs were highly diverse in their potential to express neurotrophins, SC marker proteins, as well as myelin-marker proteins. It is uncertain what features in activated SCs, SC-like cells, or naïve ASCs are most important for sufficient nerve regeneration. Possibly, the secretion of neurotrophins, such as NGF by SC-like ASCs, is crucial to initiate nerve healing, whereas the theoretically positivity for myelin markers might be a feature that is needed for the completion of nerve recovery. However, the application of ASCs in regenerative medicine is also still generally debated, and many regulatory restrictions have to be taken into account before clinical utilization [43].

## 5. Conclusions

ASCs hold great therapeutic promise in the field of nerve regeneration. The results of this experiment demonstrate that dASCs comprise the capability to mimic various functional capacities of SCs. NGF is an important regulator of axon elongation. Laminin seems to promote the NGF release of dASCs, and hence, it might be a suitable candidate for nerve-coating guidance conduits. However, the PCR data were more convincing than the observations on the protein level. A further understanding of the effects of dASC phenotypes is a prerequisite for targeted approaches to stem-cell-based nerve-injury treatment.

## Figures and Tables

**Figure 1 cells-11-02887-f001:**
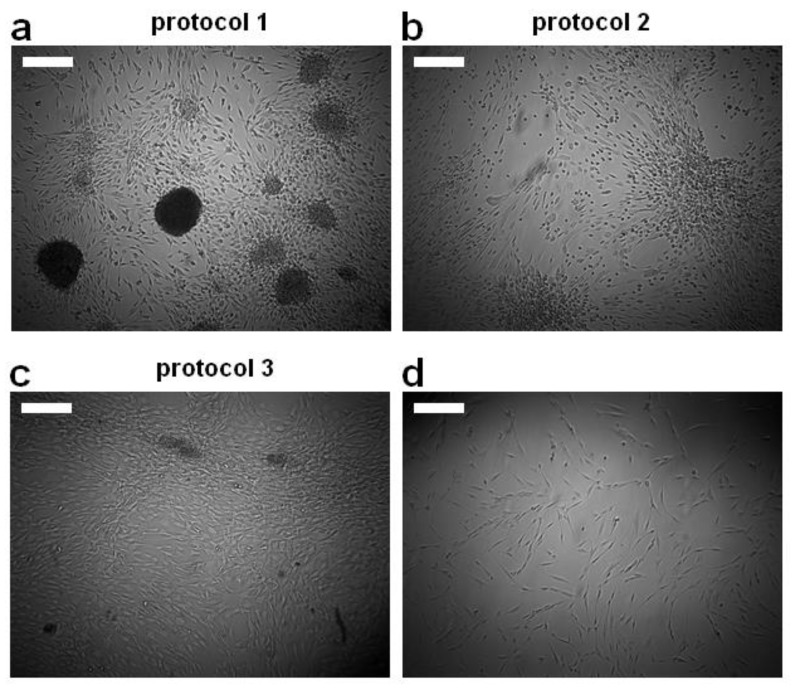
Using different glial differentiation protocols, ASCs can be differentiated into different phenotypes. (**a**) Morphological appearance of cells differentiated with protocol 1. (**b**) Morphological appearance of cells differentiated with protocol 2. (**c**) Morphological appearance of cells differentiated with protocol 3. (**d**) Morphological appearance of undifferentiated cells. Scale bars are 300 µm.

**Figure 2 cells-11-02887-f002:**
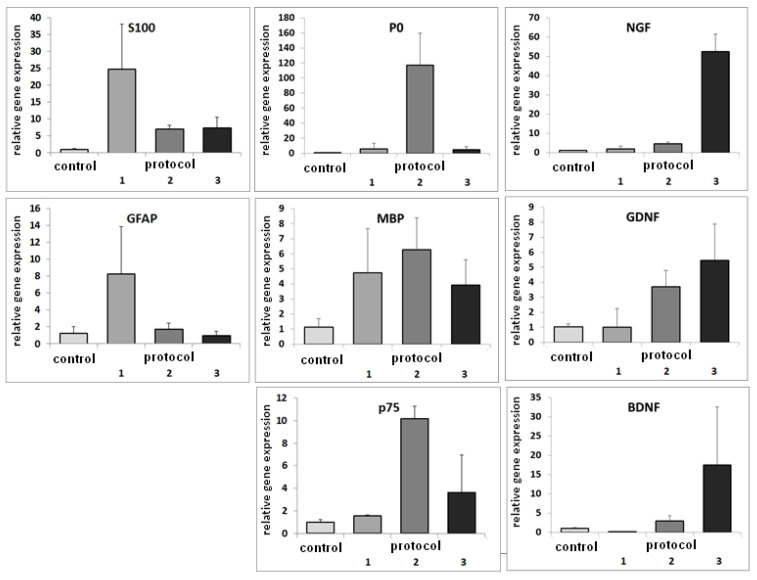
Relative gene expressions after neural differentiation of ASCs with three different protocols. ASCs differentiated with Protocol 1 express the general glial markers S100 and GFAP. ASCs differentiated with Protocol 2 express the markers of the myelinating Schwann cells P0, MBP, and p75. ASCs differentiated with Protocol 3 express the neurotrophins and neurotrophic factors NGF, GDNF, and BDNF.

**Figure 3 cells-11-02887-f003:**
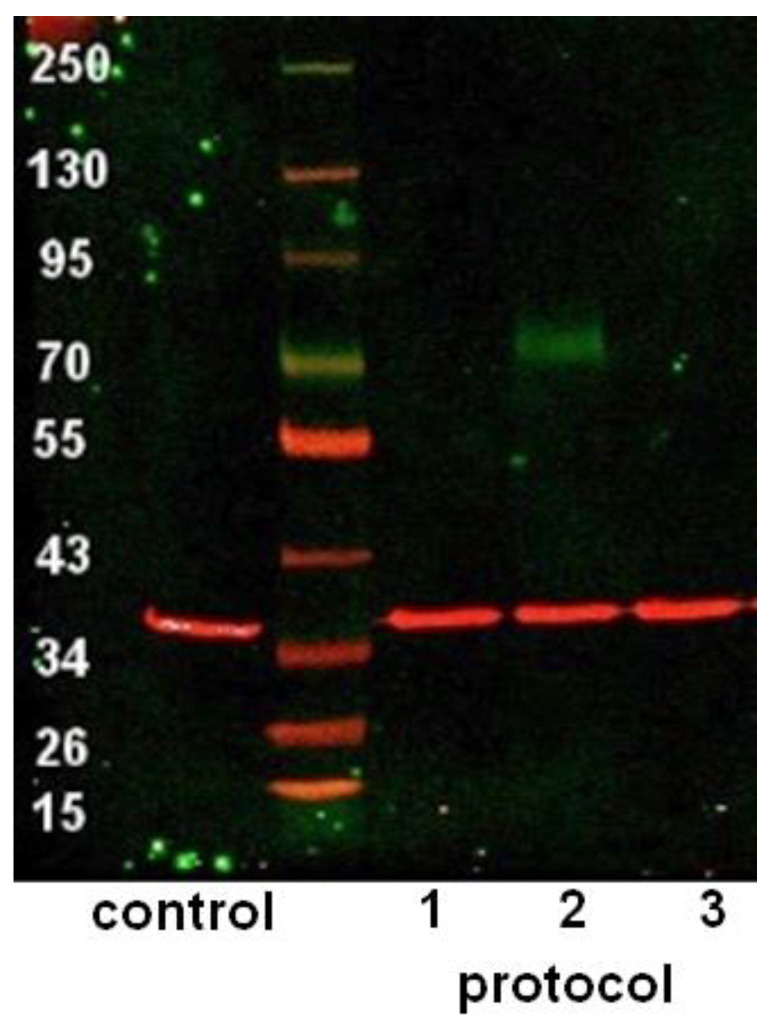
Western blot with proteins isolated from differentiated ADSCs. Expression of p75 (displayed in green) can be observed after differentiation with Protocol 2 only. GAPDH served as loading control (displayed in red).

**Figure 4 cells-11-02887-f004:**
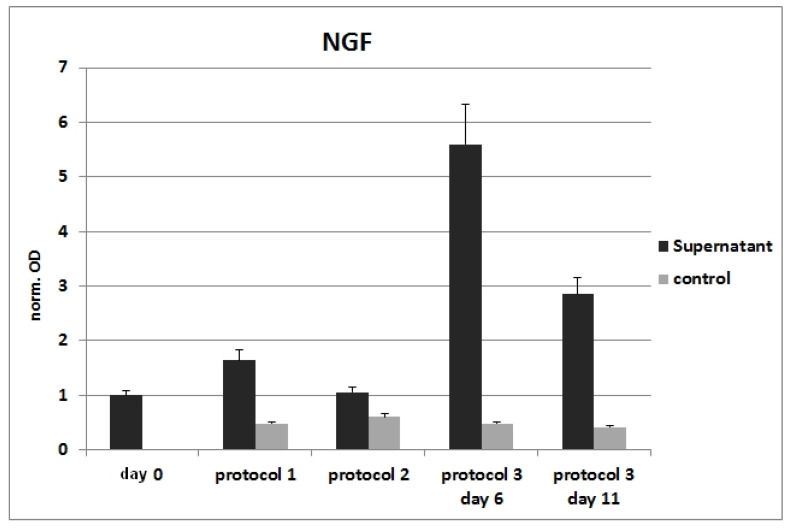
ELISA with supernatants from neuronally differentiated ADSCs. Expression of NGF is especially upregulated after differentiation with Protocol 3. The obtained values were not normalized to the total cell number.

**Figure 5 cells-11-02887-f005:**
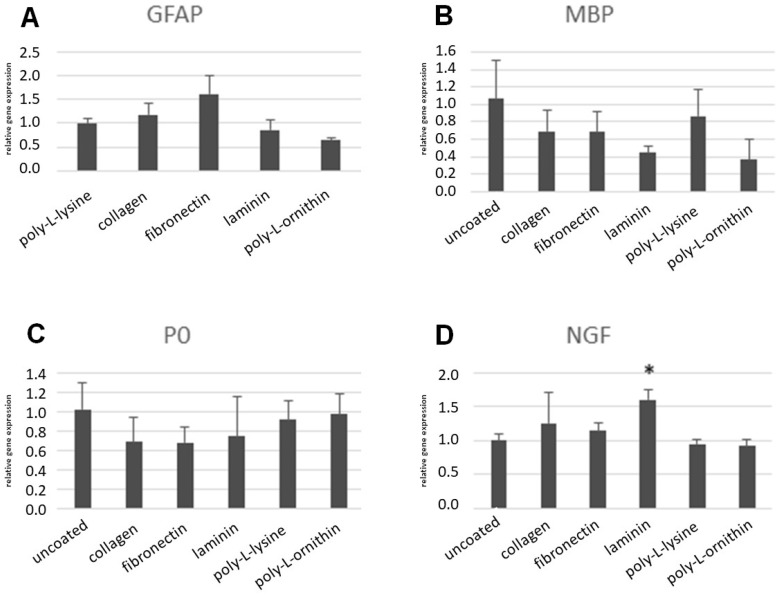
Comparison of marker gene expressions with qRT-PCR after differentiation on coated surfaces. (**A**) Expression of GFAP after differentiation with Protocol 1. (**B**,**C**) Expressions of MBP and P0 after differentiation with Protocol 2. (**D**) Expression of NGF after differentiation with Protocol 3. *: *p*-value < 0.05.

**Figure 6 cells-11-02887-f006:**
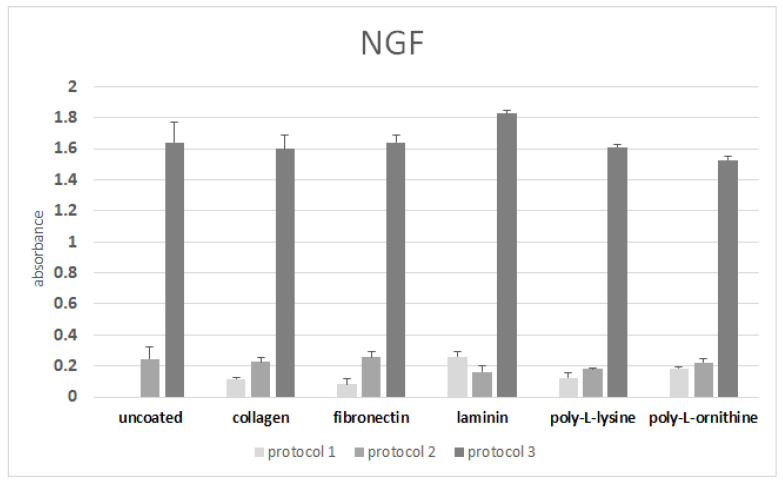
ELISA: expression of NGF after differentiation with Protocols 1, 2, and 3. The obtained values were not normalized to the total cell number.

**Figure 7 cells-11-02887-f007:**
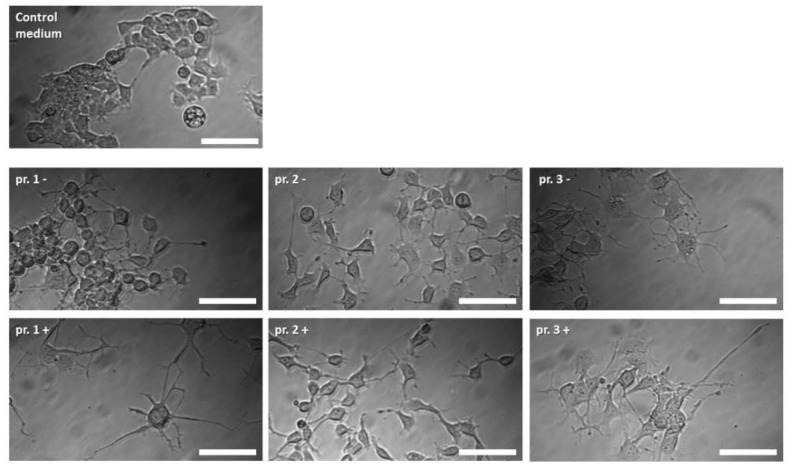
Morphological appearance of NG108-15 cells after cultivation with medium from Protocols 1, 2, and 3, with (+) and without (−) conditioning with predifferentiated ASCs. Scale bars are 100 µm.

**Figure 8 cells-11-02887-f008:**
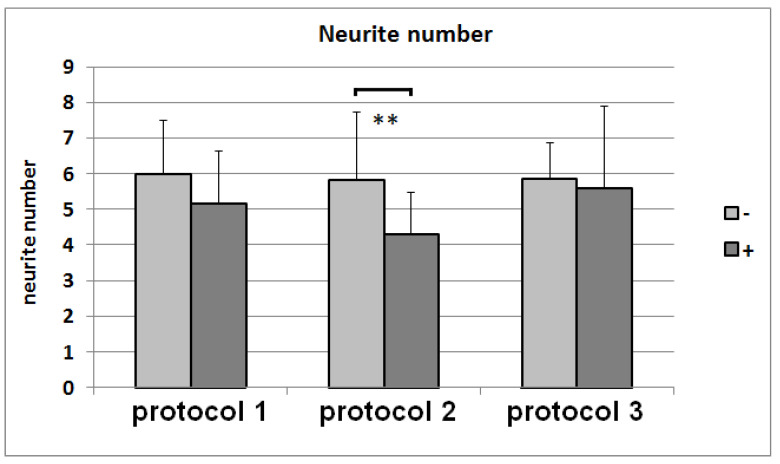
Numbers of neurites on NG108-15 cells after cultivation with media from Protocols 1, 2, and 3, with (+) and without (−) conditioning with predifferentiated ASCs. The mean value and standard deviation for each condition are shown. **: *p*-value < 0.01.

**Figure 9 cells-11-02887-f009:**
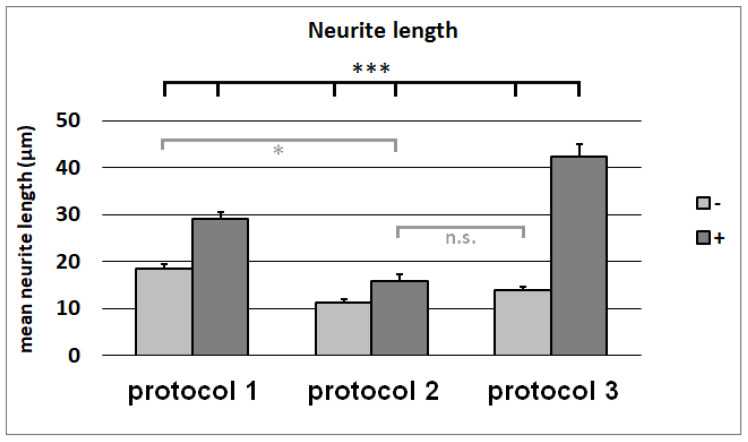
Lengths of neurites in µm on NG108-15 cells after cultivation with media from Protocols 1, 2, and 3, with (+) and without (−) conditioning with predifferentiated ASCs. The mean value and standard error of the mean for each condition are shown. The differences between every two conditions are statistically highly significant, except for Protocol 2 + NGF, which differs from Protocol 1, without NGF being statistically significant, and does not statistically significantly differ from Protocol 3 without NGF. n.s.: not significant; *: *p*-value < 0.05; ***: *p*-value < 0.001.

**Figure 10 cells-11-02887-f010:**
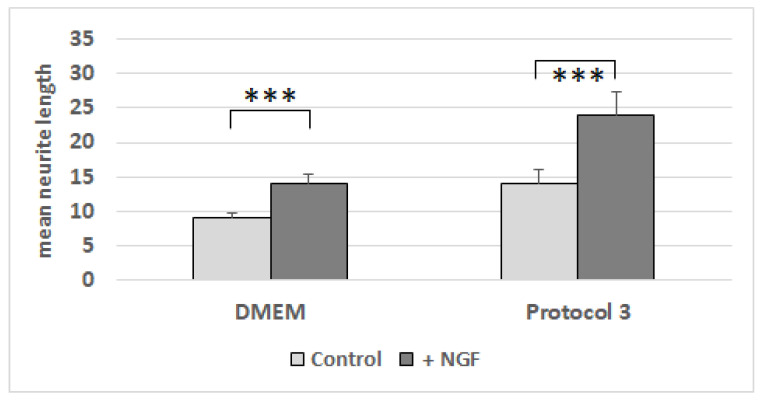
Lengths of neurites in µm on NG108-15 cells after cultivation with DMDM or Protocol-3 medium, with and without NGF supplementation. The mean value and standard error of the mean for each condition are shown. ***: *p*-value < 0.001.

**Figure 11 cells-11-02887-f011:**
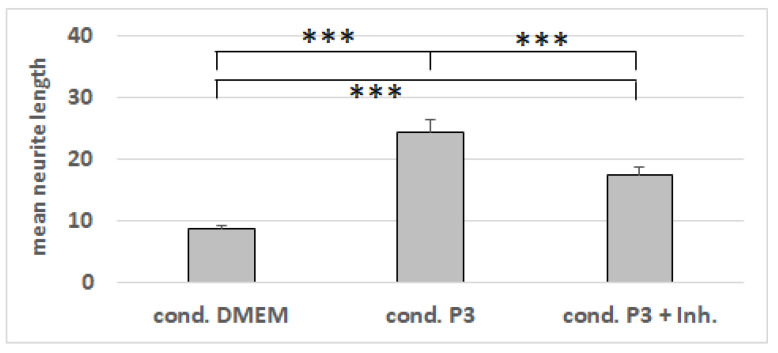
Lengths of neurites in µm on NG108-15 cells after cultivation with conditioned Protocol-3 medium, with or without NGF-neutralizing antibodies. The mean value and standard error of the mean for each condition are shown. ***: *p*-value < 0.001.

**Figure 12 cells-11-02887-f012:**
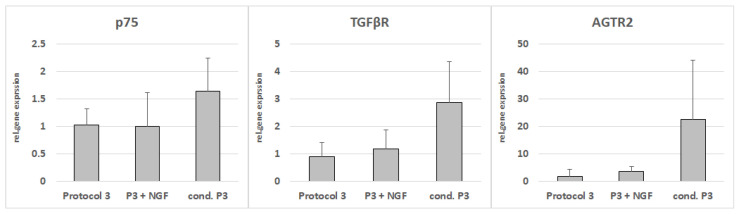
Gene expressions of p75, TGFR, and angiotensin II type 2 receptor (AGTR2) in NG-108-15 cells after indirect coculture with conditioned Protocol-3 medium, NGF-supplemented Protocol-3 medium, or unaltered Protocol-3 medium as control.

**Table 1 cells-11-02887-t001:** Compositions of the three original protocols taken from the literature, and the compounds that these protocols were supplemented with. The highlighted compound showed the strongest enhancement of the original protocol, and it was used for all further experiments. The original Protocol 1 includes two steps, and the original Protocol 3 includes three steps with different media compositions. The additional supplementation was tested for the last step of each protocol.

	Protocol 1 [13]	Protocol 2 [14]	Protocol 3 [15]
Original protocol composition	EGF bFGF ---------------------------------- all-trans-retinoic acid neuregulin 1 PDGF forskolin	N2 Supplement bFGF EGF all-trans-retinoic acid TGF-beta1	β-mercaptoethanol --------------------------------- all-trans-retinoic acid --------------------------------- PDGF bFGF forskolin neuregulin 1
Additionally tested supplements	FK-506 GAS-6 heregulin beta 1 melatonin **TGF-beta 1**	FK-506 forskolin GAS-6 heregulin beta 1 melatonin **neuregulin 1** PDGF	FK-506 GAS-6 heregulin beta 1 melatonin **TGF-beta 1**

**Table 2 cells-11-02887-t002:** Primer sequences.

Gene	Forward Primer	Reverse Primer
AGTR2	5’-GGTCTGCTGGGATTGCCTTAATG-3’	5’-ACTTGGTCACGGGTAATTCTGTTCT-3’
BDNF	5’-AATAATGTCTGACCCCAGTGCC-3’	5’-ATTGTTGTCACGCTCCTGGT-3’
GAPDH	5’-GGAGCGAGATCCCTCCAAAAT-3’	5’-GGCTGTTGTCATACTTCTCATGG-3’
GDNF	5’-CGCTGACCAGTGACTCCAATA-3’	5’-CTCTGCGACCTTTCCCTCTG-3’
GFAP	5’-GACACCTGGGTACCATTCCG-3’	5’-CATCTTGGAGCTTCTGCCTCA-3’
MBP	5’-TCTGGCAAGGACTCACACAC-3’	5’-TCTGCCTCCGTAGCCAAATC-3’
NGF	5’-GGCCACTCTGAGGTGCATAG-3’	5’-CTGTGTACGGTTCTGCCTGT-3’
P0	5’-GGCCATTGTGGTTTACACGG-3’	5’-GGAAGATTGAAATGGCATCTCGG-3’
p75	5’-AATGCGAAGAGATCCCTGGTC-3’	5’-GGAATGAGGTTGTCGGTGGT-3’
S100B	5’-AGAGGGTGACAAGCACAAGC-3’	5’-TCCTGCTCTTTGATTTCCTCCAG-3’
TGFβR	5’-ATTGGCGGAATCCACGAAGA-3’	5’-GTAGAACAACCGGCCTCCAA-3’

## Data Availability

Not applicable.

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
