# Peer review of "Peripheral Nerve Regeneration–Adipose-Tissue-Derived Stem Cells Differentiated by a Three-Step Protocol Promote Neurite Elongation via NGF Secretion"

_cells, 2022, doi:10.3390/cells11182887_

Round 1

Reviewer 1 Report

In the submitted manuscript “Stem cell based peripheral nerve regeneration :…” Klein and colleagues describe effects of conditioned media from ASCs (adipose tissue derived stem cells) pre-differentiated with use of 3 distinct culture protocols on neurite outgrowth in neuron-like cells. ASCs are stem cells of mesodermal origin, however upon cultivation in particular culture conditions they start to express markers of Schwann cells, secrete neurotrophins, and aid the regeneration of peripheral nerves. The current study compares the expression of astrocytic and Schwann cell markers in induced ASCs in vitro, as well as it measures various cytokines expression and secretion. Authors demonstrate that one of the protocols (#3) triggers NGF secretion, and suggest that differentiated ASCs-derived NGF promotes the extension of neurite-like processes in neuron-like cell culture.

The topic of the study is of significant importance for the field of peripheral nerve regeneration. The presented data help to identify the potential of manipulations performed on ASCs in order to supplement/replace Schwann cells function during PNS regeneration. The complexity of regeneration process, translating into the requirements differentiated ASCs must fulfil to enable the regeneration is well described in the Discussion chapter.

However, there are some technical flows and inconsistencies in the manuscript that need to be addressed to fully validate the observations made by Authors.

MAJOR POINTS:

1.       The information on statistical methods used in the study is not provided. The number of repeats is not known. Please provide the missing information, both at the Materials and Methods part, and number of repeats, statistical test type, significance at the Results chapter.

2.       When cytokines are measured with ELISA, the obtained values should be normalized to total cell number. The increased level of NGF in conditioned medium may result from greater cell density upon particular protocol application.

3.       The composition of each of 3 protocols media is not very clearly listed. First, Authors introduce basic protocols. Next, many other components (neuregulin, TGFb) are mentioned, followed by the information that sometimes they are added, sometimes they are not. Please provide the final list of components of each differentiating medium at once (basic protocol as published in the cited paper and modification introduced by Authors).

4.       Authors claim that ASCs seeding on laminin coated plates potentiates protocol #3 dependent NGF secretion, actually seen at the transcriptional level. Furthermore, the role of laminin coating is emphasised in the manuscript title. However, when NGF amount in conditioned medium is evaluated (protein level, Figure 6), this effect cannot be observed. Therefore, reformulating the conclusions and the title of manuscript is strongly recommended in order to minimize reader’s confusion.

5.       Figure 11. Please provide the information how many times the experiment was repeated, and what number of neurite-like processes was analysed per one experiment. First of all, given the error bar size, it is hard to acknowledge the difference between the protocol #3 conditioned medium effect and medium plus inhibitor effect on neurite length. On the other hand, comparison of the bar#1 and bar#3 (about 2-fold difference) leads to the conclusion that a) NGF signalling blockade is not very efficient b) some other factors in medium preconditioned along protocol #3 greatly contribute to neurite elongation. Please supplement the missing information and comment on the data.

6.       Figure 12. Unknown number of repeats, statistical significance not determined, here one may notice that NGF doesn’t upregulate the expression of investigated genes, whereas protocol#3 induced ASCs medium does. There is no explanation or discussion offered in the manuscript. Please justify the presence of these data in the manuscript or remove them to avoid readers confusion.

7.       Introduction: it would be interesting to provide more information on guidance conduits. What are their composition and effects on nerve regeneration? Especially, when significant portion of the paper describes effect of ECM elements used for coating on pre-differentiated ASCs properties.

MINOR POINTS:

8.       Neuroblastoma cells and differentiated ASCs are not co-cultured.

9.       The abbreviations should be developed right after they are used in the text (like SC-like cells in the abstract, gene names etc.).

10.   RT-PCR data are presented (text) in very awkward way, what actually means that “relative gene expression of S100B is 25”? The title of y axis in Figure 2 and others needs to be corrected/added.

11.   Line 235. NG108-15 cells shouldn’t be defined as neuronal cells, it is safe to use “neuron-like cells” instead.

12.   Neurite length needs to be expressed in micrometres rather than pixels. 

1.       Line 60: Cell Culture subchapter: there are two cell types used in the study: ASCs and NG108-15, this subchapter needs to be divided accordingly. Moreover, the coating used for NG108-15 cells culture needs to be specified. 

c

Reviewer 2 Report

The work presented involves a comparison of three different protocols to differentiate adipose-derived stem cells for their use in nerve regeneration in a manner like Schwann cells. Nerve guidance conduits were chosen as a test substrate to demonstrate the efficacy of the different approaches. The work is interesting and has presented some interesting findings. A few questions and comments are provided to help the authors improve the presentation of their work

1. Minor language editing is recommended.

2. In the materials and methods section, 37O is used. Is this in Centigrade or Fahrenheit?

3. The absence of scale bars in the images of cells makes it difficult to assess. For example, Fig. 1 is taken using 40x magnification. However, the images are more like 4x. Scale bars would eliminate such questions.

4. In the protocol for Western Blot, the ladder used is not mentioned. Is this the Chameleon duo or a different ladder?

5. For determining the neurite growth, how many images were analyzed? How was the average length calculated? What do the error bars in the graphs represent?

6. In Fig. 9, the neurite length in response to pr3+ appears high. However, from the images presented in Fig. 7, pr1+ appears to have more cells with longer neurites than pr3+. What were the criteria for measuring neurite length? How was the comparison made?

7. Works from Mazini et al. Stem Cell Research & Therapy, 2021; Zhou, L.N et al. Cell Res Ther 2020; Wu et al., Front Cell Dev Biol. 2021 Apr 30; 9:658099; Urrutia et al. PLOS ONE, 2019 would be good additions to references and may even provide some information to consider towards the focus of the manuscript.

Overall, the work is interesting. However, the novelty of the work is not apparent. The reason for using three different protocols for cell differentiation and the outcome is unclear.

Reviewer 3 Report

Dear Authors 

I congratulate your work on exploring the regenerative capacity of the ASCs in neuronal damage to compensate for the SCs. The experiments were well planned and executed. I would like to know the basis behind the selection of components in the differentiation protocols #2, #3 while the rationale for protocol #1 is discussed and also the choice of growth factors utilized for the differentiation of ASCs. 

Round 2

Reviewer 1 Report

Most issues were sufficiently addressed in Authors response and revised manuscript. Still, I am not convinced why effect of laminin coating on NGF expression/secretion is so important.  Unfortunately, this experiment (Figure 6, and expt. shown in Figure 4) lacks proper normalization, and together with very small increase in NGF secretion, is a main drawback in this part of the overall interesting story. Here, rather NGF secreted by protocol #3 pre-differentiated ASC’s (regardless of the coating used) seems to be a central figure in the manuscript. The role of ASC’s produced factors should be emphasized in the title, with special focus on NGF secretion, and its effect on neuronal-like cells differentiation. Laminin effect on NGF gene expression is just additional observation, especially without sufficient evidence at the protein level.

Figures 8 & 9 – Are the data statistically significant? If yes, please provide p-value.

Line 230 – if cells are differentiated toward astrocytes or Schwann cells, “neuronally” is not appropriate

In the case of large sample cohort, standard deviation error bars may be replaced with standard error bars (SE or SEM) - Figures showing neurite number/length.

The Discussion part very adequately addresses all issues arising during results analysis. Only the sentence in lines 375-377 needs to be reformulated, here the word “release” needs to be replaced with “production” or “expression”.

Author Response

Comments and Suggestions for Authors

Most issues were sufficiently addressed in Authors response and revised manuscript. Still, I am not convinced why effect of laminin coating on NGF expression/secretion is so important.  Unfortunately, this experiment (Figure 6, and expt. shown in Figure 4) lacks proper normalization, and together with very small increase in NGF secretion, is a main drawback in this part of the overall interesting story. Here, rather NGF secreted by protocol #3 pre-differentiated ASC’s (regardless of the coating used) seems to be a central figure in the manuscript. The role of ASC’s produced factors should be emphasized in the title, with special focus on NGF secretion, and its effect on neuronal-like cells differentiation. Laminin effect on NGF gene expression is just additional observation, especially without sufficient evidence at the protein level.

Response: We thank the reviewer for the critical remarks. We agree that the effect of laminin coating is not distinctive enough on the translational level to justify the title. Frankly, with the PCR data in mind, we wanted to have a somewhat catchy title with a clear takeaway message. However, we have to agree with the reviewer that the title is not supported by the data. Therefore, we provide a new title for our manuscript:

Peripheral nerve regeneration - Adipose tissue derived stem cells differentiated by a three-step-protocol promote neurite elongation via NGF secretion

Figures 8 & 9 – Are the data statistically significant? If yes, please provide p-value.

Response: We have provided the information about statistical significance for figure 8 and 9.

Line 230 – if cells are differentiated toward astrocytes or Schwann cells, “neuronally” is not appropriate

Response: The reviewer is correct, of course. We simply overlooked this mistake in the figure legend. We erased the word.

In the case of large sample cohort, standard deviation error bars may be replaced with standard error bars (SE or SEM) - Figures showing neurite number/length.

Response: We thank the reviewer for the suggestion. For neurite length diagrams (figures 9, 10, and 11), we have changed the standard deviation to standard error (of the mean). However, we left the standard deviation for figure 8 (neurite number), because here, the sample size is lower by the factor ~5 (the average neurite number per cell, for neurite length, every single neurite measurement is a sample, but for neurite numbers only every cell (with ~5 neurites each) is a sample).

The Discussion part very adequately addresses all issues arising during results analysis. Only the sentence in lines 375-377 needs to be reformulated, here the word “release” needs to be replaced with “production” or “expression”.

Response: We want to thank the reviewer for the attentive review and apologize for our mistake. Although we have rephrased the discussion to direct the focus away from the protein level, we missed that “release” clearly indicates exactly this. We have replaced “release” with “transcription”.